# Micro triboelectric ultrasonic device for acoustic energy transfer and signal communication

Chen Chen [1,2], Zhen Wen [1✉], Jihong Shi[1], Xiaohua Jian[3], Peiyang Li[3], John T. W. Yeow [2✉] & Xuhui Sun [1✉]

As a promising energy converter, the requirement for miniaturization and high-accuracy of triboelectric nanogenerators always remains urgent. In this work, a micro triboelectric ultrasonic device was developed by integrating a triboelectric nanogenerator and micro-electro-mechanical systems technology. To date, it sets a world record for the smallest triboelectric device, with a 50 μm-sized diaphragm, and enables the working frequency to be brought to megahertz. This dramatically improves the miniaturization and chip integration of the triboelectric nanogenerator. With 63 kPa@1 MHz ultrasound input, the micro triboelectric ultrasonic device can generate the voltage signal of 16.8 mV and 12.7 mV through oil and sound-attenuation medium, respectively. It also achieved the signal-to-ratio of 20.54 dB and exhibited the practical potential for signal communication by modulating the incident ultrasound. Finally, detailed optimization approaches have also been proposed to further improve the output power of the micro triboelectric ultrasonic device.

[1] Institute of Functional Nano and Soft Materials (FUNSOM), Jiangsu Key Laboratory for Carbon-Based Functional Materials and Devices, and Joint International Research Laboratory of Carbon-Based Functional Materials and Devices, Soochow University, No. 199 Ren-ai Road Suzhou Industry Park, 215123 Suzhou, China. [2] Advanced Micro-/Nano-Devices Lab, Department of Systems Design Engineering, Waterloo Institute for Nanotechnology, University of Waterloo, 200 University Avenue West, Waterloo, ON N2L 3G1, Canada. [3] Suzhou Institute of Biomedical Engineering and Technology, Chinese Academy of Sciences, No. 88 Keling Road Suzhou New District, 215163 Suzhou, China. ✉email: wenzhen2011@suda.edu.cn; jyeow@uwaterloo.ca; xhsun@suda.edu.cn

Acoustic energy transfer (AET) has becoming an attracting topic in low-power energy transfer[1]. Up till now, most reported AET systems are based on the bulk lead zirconate titanate (PZT) transducers with size ranging from millimeters to centimeters[2,3]. The birth of triboelectric nanogenerator (TENG), an emerging technology for converting mechanical energy to electricity, provides an alternative solution[4–8]. Inspired by the simple configuration, more material choices and specific output, diverse demonstrations of TENG have been proposed over last decade, such as implantable applications[9–13]. Although it has been reported to directly harness the mechanical energy inside human body, it suffers from the limitations of implanted site, uncertain output power and poor chip integration[14–17]. Several efforts have been devoted to developing self-powered triboelectric acoustic sensor (TAS)[18–21]. Most TASs can only collect audible acoustic signal, whose frequency typically ranges from hundreds of to thousands of Hertz. However, AET for powering implanted devices usually requires the frequency in the ultrasonic range. Most ultrasonic transducers have the best performance when working at resonance mode[22,23]. Since all the reported TASs are membrane-based, the device sizes approached sub-millimeter when resonant frequency goes to ultrasonic range. As a result, conventional fabrication methods can hardly meet the requirements.

Chip-sized integration is another important issue limiting the development of TAS. Acoustic device is a complicated hybrid via integration among different systems, including sensor and circuit. The whole system will be dramatically miniaturized if we could integrate everything on a single chip. To date, most reported TAS are bulky and none of them has addressed this issue yet. Micro-Electro-Mechanical Systems (MEMS) technology as one of the most promising technologies has revolutionized industry in many aspects[24,25]. By advanced processing and patterning techniques, sophisticated structures can be easily manufactured ranging in size from a few micrometers to millimeters. Over last decades, various MEMS acoustic sensors with different working mechanisms have

been proposed[26–28]. Benefited from this technology, MEMS acoustic sensors have the advantages of miniaturization, batch manufacturing process and easer integration with other electronic systems. Yet, the integration of TENG and MEMS device to solve miniaturization and high-accuracy problems still pose challenges.

In this work, a microstructured triboelectric ultrasonic device (μTUD) was developed based on the coupling technologies of TENG and MEMS. Silicon and silicon oxide respectively acted as the triboelectric pair and a vacuum cavity was built to eliminate the negative effect of the ambient environment. To our best knowledge, this is the very first time that TENG has been fabricated by MEMS process and brought into the microscale. The working frequency of the μTUD is 1.17 MHz, which is toward ultrasonic range without precedent. Moreover, AET through oil and sound-attenuation medium were demonstrated and 16.8 mV and 12.7 mV output voltage were achieved respectively. Pulse experiments were also carried out and a signal-to-noise ratio (SNR) of 20.54 dB was achieved, indicating that the signals can even be modulated from the transmitting end. It is worth noting that the energy conversion efficiency of the μTUD can potentially reach up to ~33% by theoretical analysis. Most importantly, this work greatly promotes the miniaturization and integration level of TENG and exhibits a new pathway for triboelectric device design.

## Results

### Structure design and working mechanism of the μTUD.
Essentially, the basic structure of the μTUD is a micro capacitor, where a fully clamped silicon membrane is suspended atop a vacuum cavity (Fig. 1a). Highly doped silicon layer underneath the silicon oxide (triboelectric & dielectric layer) acts as bottom electrode. The silicon membrane is also highly conductive so that the combination of silicon membrane and top gold layer functions as top electrode. Fig. 1b, c illustrates the basic structure and the working mode (conductor-dielectric contact-separation mode) of

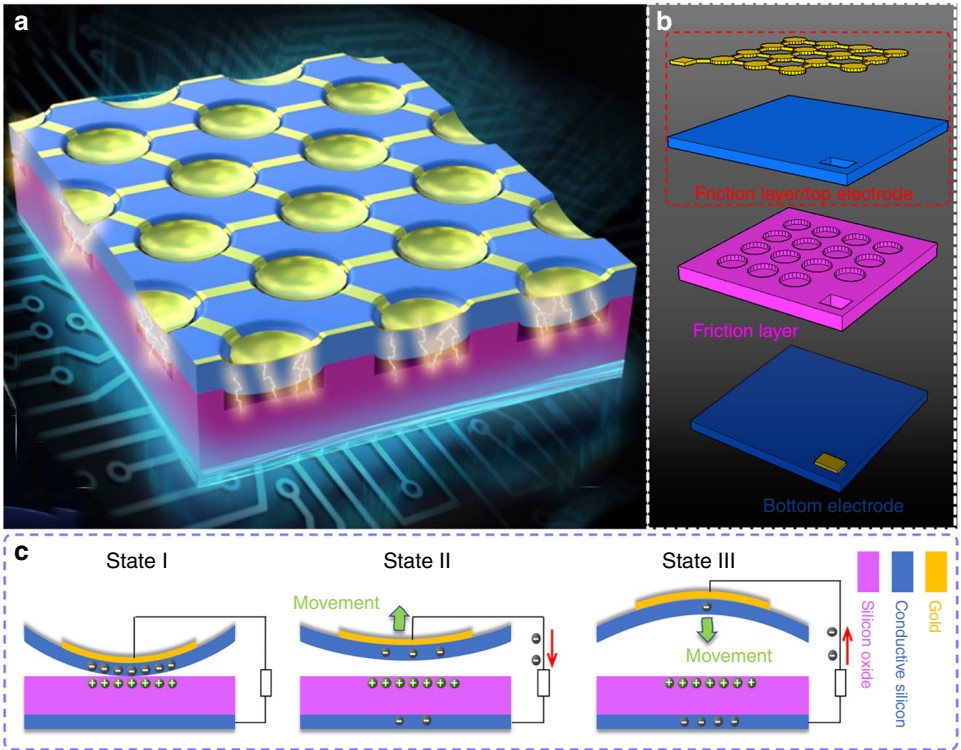

**Fig. 1 Structure illustration and working mechanism of the micro triboelectric ultrasonic device (μTUD). a** Representative diagram of the μTUD. **b** Exploded view structure of the μTUD. **c** Schematic of the working mechanism of the μTUD during vibration.

the μTUD. The suspended silicon membrane vibrates and contacts the silicon oxide layer under the excitation of the incident ultrasound. Then, the contact between silicon and silicon oxide causes the charge redistribution by triboelectrification[29,30]. Scanning Kelvin probe microscopy (SKPM) was performed on the surface of silicon oxide to obtain the surface potentials. The surface potentials before and after contacting are shown in Supplementary Fig. 1. The surface potential after contact with polytetrafluoroethylene (PTFE) film was also measured as a reference. From the results, the surface potential of silicon oxide presents obvious increase after contact, indicating an accumulation of positive charges. When this contact-separation process repeats, AC power is generated in the external circuit. Although more significant triboelectrification between PTFE and silicon oxide is noticed, tradeoff should be made between the output performance and fabrication limit of MEMS process. To maximize the output power, hundreds of cells can be fabricated in parallel within one single chip.

Based on the working mechanism, two aspects should be cautiously considered in the structure design of the μTUD: resonant frequency and cavity depth. For this diaphragm-structured μTUD, the maximal output will be achieved at the 1st mode resonance and the resonant frequency mainly depends on the size of the diaphragm.

In medium, acoustic attenuation can be modeled as

$$I_Z = I_0 e^{-\alpha Z}, \tag{1}$$

where $I_Z$ is the intensity of ultrasound at a certain depth ($Z$), $I_0$ is the original intensity of ultrasound, and $\alpha$ is the attenuation of medium, respectively. Therein, $\alpha$ is linearly dependent on the frequency of the ultrasound[31].

Meanwhile, some side-effects may also occur when the ultrasound travels through human tissues, such as cavitation. To estimate the mechanical damage, mechanical index (MI) is often specified, which is defined as[32]

$$MI = \frac{p}{\sqrt{f_0}} \cdot \frac{\sqrt{1\,MHz}}{1\,MPa}, \tag{2}$$

where $p$ is the peak negative ultrasound pressure in MPa derated by $0.3\,dB\,cm^{-1}\,MHz^{-1}$ and $f_0$ is the center frequency of ultrasound in MHz. Recommended by U.S. Food and Drug Administration (FDA), the MI for the ultrasound diagnostic system should not be higher than 1.9. Combining Eqs. (1) and (2), it is obvious that higher frequency will result in a heavier attenuation while lower frequency will induce a large MI value. Several studies have explored that the suitable range of working frequency of AET for bio-applications is 0.5–2 MHz[33–35].

A novel μTUD with circular membrane, whose cross-sectional schematic is depicted in Fig. 2a, was developed based on MEMS technology. As aforementioned, the resonant frequency of the μTUD is dependent on the structure of the membrane. We modeled the membrane as a circular plate with all edges clamped. Since degassed/deionized water has similar acoustic properties to tissues (Supplementary Table 1), we applied water-loaded boundary condition to one side of the plate. Additionally, the effect of the top gold layer is eliminated since the thickness of the gold layer is much smaller than that of silicon membrane. The 1st mode resonant frequency of the circular membrane in water is[36]

$$f = \frac{0.474 \frac{hc_p}{R^2}}{\left(1 + 0.67 \frac{\rho_w R}{\rho h}\right)^{1/2}}, \tag{3}$$

$$c_p = \left[\frac{E}{(1-\nu^2)\rho}\right]^{1/2}, \tag{4}$$

where $R$ is the radius of the membrane, $h$ is the membrane

thickness, $\rho$ is the volumetric density of the membrane, $E$ is Young's modulus of the membrane material, $\nu$ is Poisson's ratio of the membrane material, and $\rho_w$ is the density of water.

Other than the resonant frequency, another key point for designing the μTUD is the cavity depth. Sealed vacuum cavity was fabricated by wafer bonding process to eliminate the effect of ambient environment. In reality, the membrane is naturally curved down to the substrate due to the atmospheric pressure outside the vacuum cavity. Therefore, the membrane will initially collapse if the cavity depth is too small while the membrane will not touch the substrate if the cavity depth is too large. Cavity depth should be conscientiously chosen to satisfy the above-mentioned requirements.

On the basis of the thin plate theory, the static displacement of a circular membrane is governed by the equation of equilibrium[37]

$$\frac{d^3w}{dr^3} + \frac{1}{r}\frac{d^2w}{dr^2} - \frac{1}{r^2}\frac{dw}{dr} = \frac{Q}{D}, \tag{5}$$

where $w$ and $Q$ are the static displacement and the shearing force at a distance ($r$) away from the center of the membrane in cylindrical coordinate. $D$ is the membrane's flexural rigidity, defined as

$$D = \frac{Eh^3}{12(1-\nu^2)}. \tag{6}$$

Detailed boundary conditions and derivation process of the circular membrane are presented in Supplementary Note 1. For further verification, numerical analysis (COMSOL Multiphysics) was also utilized to simulate the resonant frequency and the static displacements for the membrane (Fig. 2c, d). Supplementary Table 2 provides the required parameters for theoretical analysis.

The results of the theoretical analysis are presented in Supplementary Table 3. The resonant frequencies of the membrane are 1.18 MHz and 1.19 MHz from analytical modeling and numerical analysis, respectively. It should be noted that the resonant frequency will slightly increase if we eliminate the top gold layer. This discrepancy is because the addition of the top gold layer decreases the effective Young's modulus of the membrane.

Based on the results of the analytical modeling, 3D mapping of the static displacement surface was plotted for the membrane as shown in Fig. 2b. Meanwhile, the static displacement surface of the membrane under the atmospheric pressure was simulated by COMSOL and the result is illustrated in Fig. 2d. Due to the computational capacity, only one quarter geometry was built and symmetric conditions were applied. Ignoring the top gold layer, the maximal displacements of the membrane under the atmospheric pressure are respectively 79.4 nm and 79.1 nm via two methods. In practice, the cavity depth should be larger than the maximal static displacement to avoid collapse. Meantime, a smaller cavity depth is desired to achieve a larger contact area. Therefore, the cavity depth is targeted to 90 nm, which is a little larger than the theoretically maximal membrane displacement.

**Fundamental characterizations of the μTUD.** To satisfy the abovementioned requirements, the cavity depth was targeted to 90 ± 10 nm in consideration of the fabrication error. The detailed fabrication process is illustrated in Fig. 3a and described in detail in experimental section. In the fabrication of the μTUD, the wafer bonding process was utilized to build vacuum cavity between the triboelectric pair. As a result, the triboelectrification will not be affected by the ambient environment and the output of the μTUD will be more stable and reliable. The optical micrographs (Fig. 3b, c) show the dimension structures of the μTUD. SEM images were also taken to show the cross-sectional structure of the μTUD in

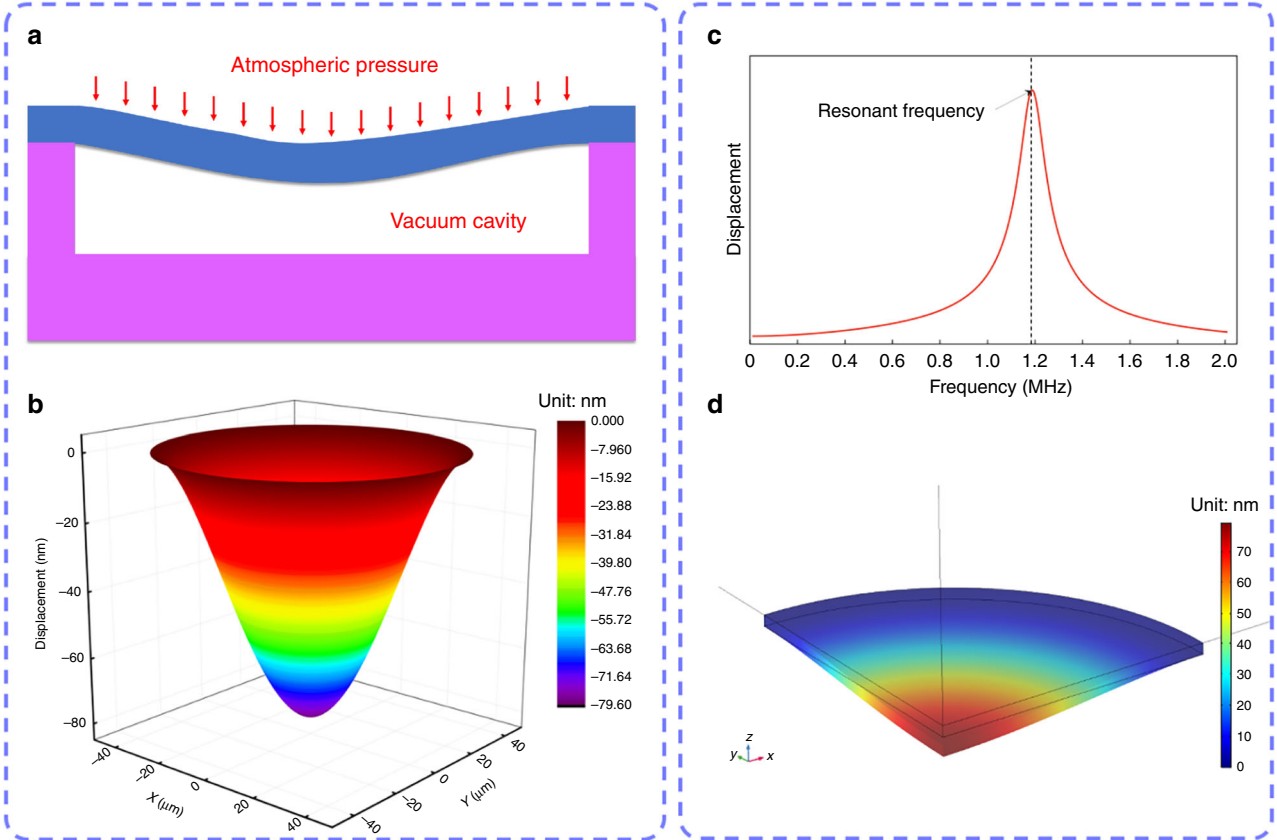

**Fig. 2 Structure design and theoretical simulation of the µTUD. a** Cross-sectional schematic of the µTUD under atmospheric pressure for analytical modeling. **b** 3D mapping of the static displacement surface of the membrane based on analytical modeling. **c** Displacement response of the µTUD in the frequency spectrum via numerical analysis. **d** Static displacement surface for the membrane based on numerical analysis.

Fig. 3d, e and it is observed that the actual cavity depth is exactly $90 \pm 10$ nm.

To characterize the electrical output of the µTUD, an ultrasound link was established. The schematic of the experimental setup is illustrated as Fig. 4a. A commercial ultrasound (US) transducer, actuated by a signal generator, was used to produce ultrasound. Both the µTUD and the US transducer were immersed into vegetable oil with a distance of ~30 mm. In the experiment, vegetable oil is normally used as medium because it has similar acoustic properties as water but can prevent shorting. During the experiments, the US transducer was actuated by a continuous wave with the amplitude of 5 V and the frequency of 1 MHz. The output open-circuit voltage of the µTUD is shown in Fig. 4b, where the maximal value is $16.8$ mV$_{P-P}$. Furthermore, a linear relationship between the peak open-circuit voltage and the incident acoustic pressure for the µTUD was also established in Fig. 4c. The linear property of the output voltage can be explained by the change of the contact area. As the incident acoustic pressure increases, the contact area between the silicon membrane and the silicon oxide substrate also increases. This induces a more sufficient triboelectrification (higher surface charge density) and a higher output voltage. In addition, the standing wave effect also affect the output of the device. The variation of the output voltage along the axial direction was observed (Fig. 4d). This phenomenon is due to the standing wave effect (Supporting Note 2). Since US transducer and the µTUD are well aligned in the AET, the standing wave may be generated between the transmitter and receiver. Considering the acoustic impedance mismatch between transducers and medium, acoustic wave reflects every time while it hits the transducer. These reflections make the acoustic energy persistent in the form of the

standing wave. Therefore, the efficiency of the ultrasonic link will be sensitive to the separation between transmitter and receiver. Only when the separation is integer multiples of one-half wavelength, the efficiency of the ultrasound link will be maximized.

**Demonstrations for AET and signal communication**. To demonstrate the application of the µTUD for AET, a similar ultrasound link was established, where same US transducer was used as external acoustic source (representative schematic shown as Supplementary Fig. 2a). The µTUD, mounted on a customized printed circuit board (PCB), functions as a receiver to harvest the acoustic energy. The load resistor was connected in series with the µTUD to calculate the output electrical power. The US transducer and the µTUD (with a distance of ~30 mm to eliminate the negative effect of standing wave) were well aligned and immersed into vegetable oil preventing from shorting (shown in Fig. 5a). A photograph of the experimental setup is shown in Supplementary Fig. 2b. A continuous sinusoidal wave (10 V$_{P-P}$, 1 MHz), produced by a signal generator, was input into US transducer and the output voltage was measured by an oscilloscope. The dependence of the output voltage and output power on the load resistance is illustrated in Fig. 5c. It is clearly seen that three regions (marked as region I, II, and III) can be defined though the output voltage keeps increasing along all three regions. The output voltage increases gently in region I and III but rapidly in region II. It can be explained that the inherent impedance of the µTUD is dominant compared to the external load resistance in region I and the circuit can be considered as short circuit. In region III, the load resistance is much larger than the inherent impedance of

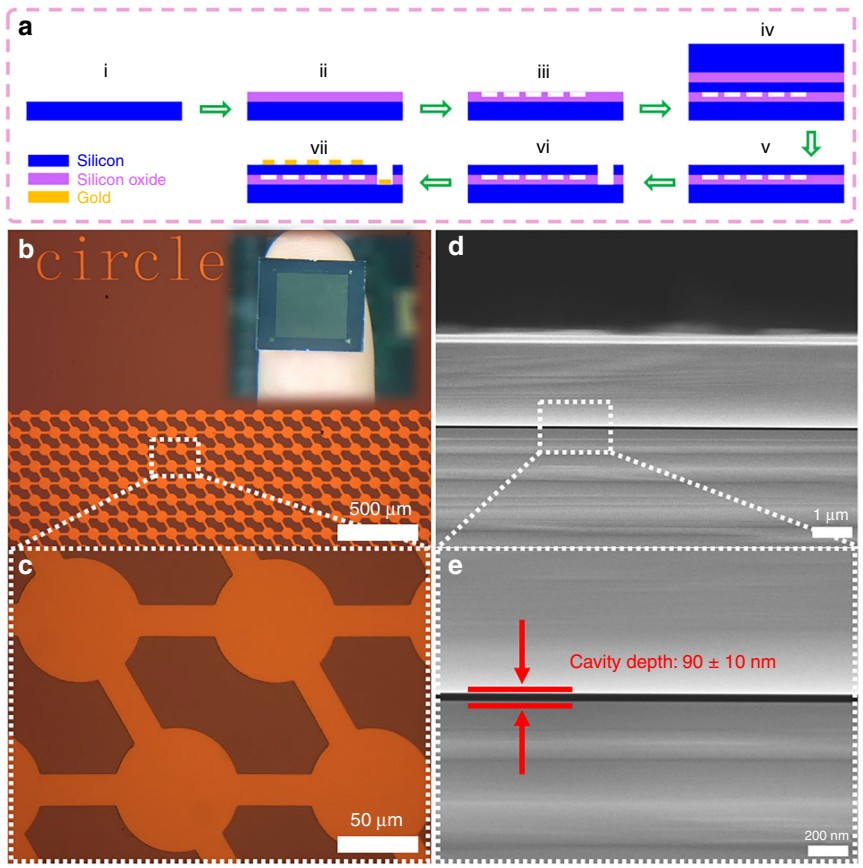

**Fig. 3 Fabrication and dimension structures of the μTUD. a** Detailed fabrication process of the μTUD. **b** Optical microscope image and (**c**) selected area enlarged image of the μTUD cells (inset shows a digital photograph). **d** SEM and (**e**) enlarged images of the cross-sectional view of the μTUD cavity.

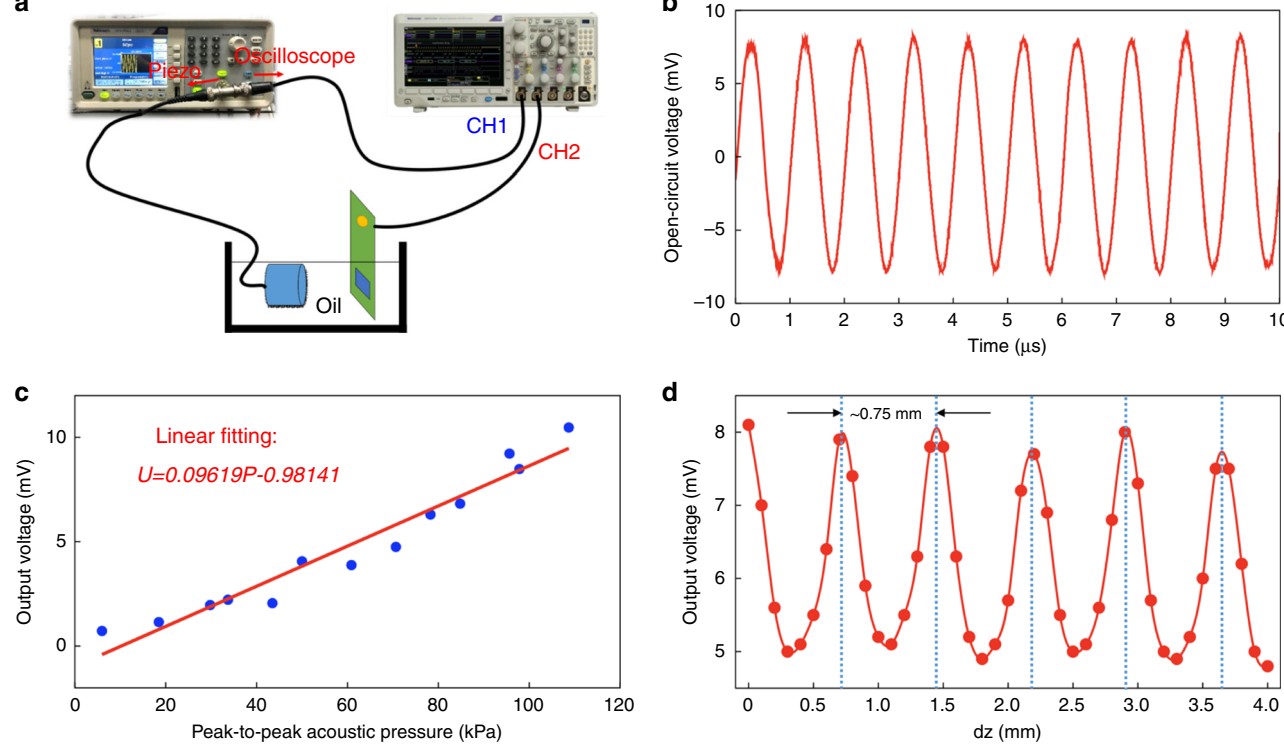

**Fig. 4 Electrical characterizations of the μTUD. a** Schematic of the experimental setup. **b** Open-circuit voltage of the μTUD. (63 kPa@1 MHz). **c** Relationship between the peak open-circuit voltage and the incident acoustic pressure for the μTUD. **d** Standing wave effect in the experiments.

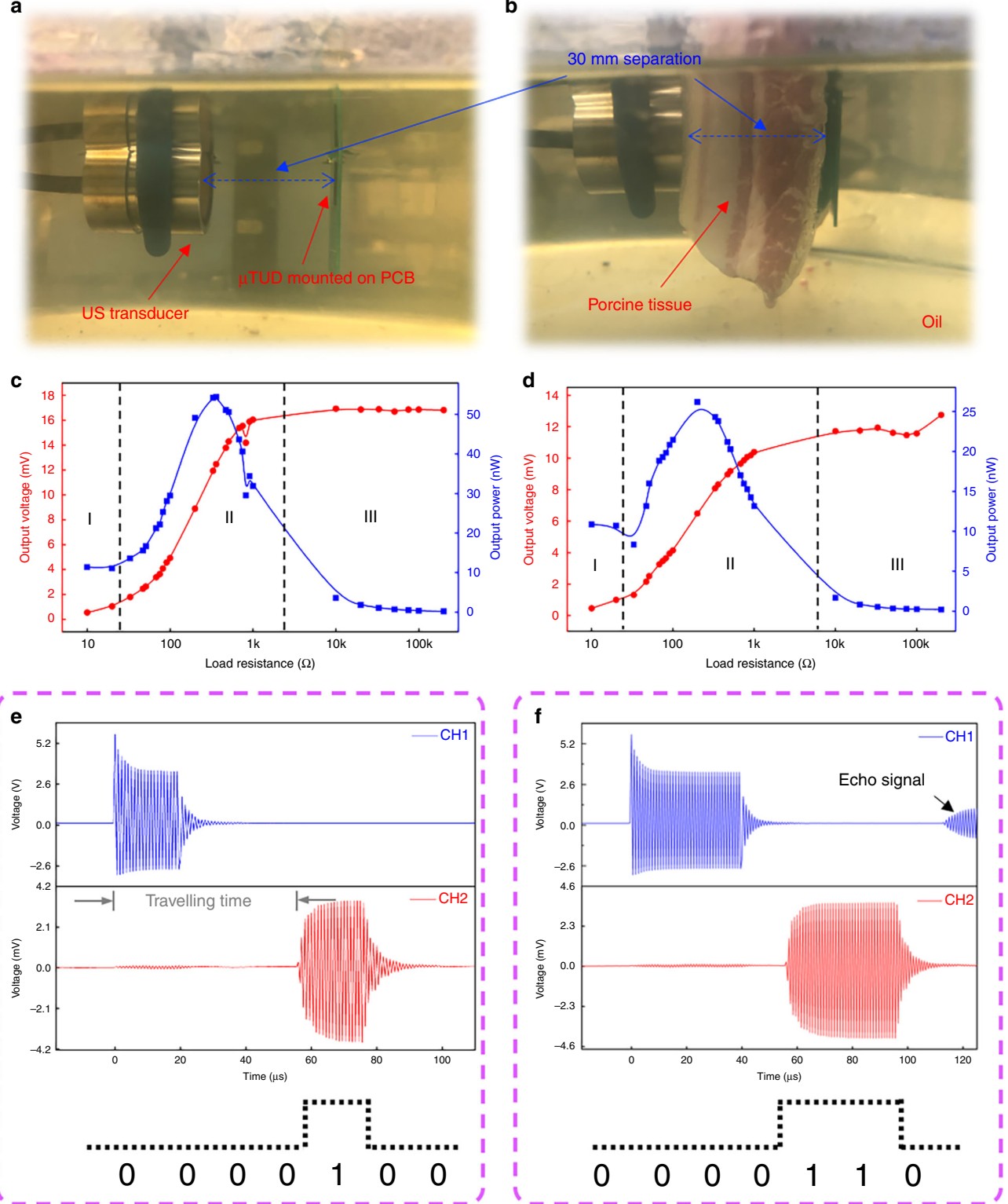

**Fig. 5 Demonstrations of the μTUD for acoustic energy transfer and signal communication.** Photograph of the ultrasonic link (**a**) without and (**b**) with sound-attenuation medium (porcine tissue). Dependence of the output voltage and the output power on the load resistor (**c**) through oil and (**d**) through sound-attenuation medium. Input signals to ultrasound transducer and the output signal from the μTUD driven by (**e**) 20-cyc and (**f**) 40-cyc sine waves (1 MHz, ~85 mm separation in oil).

the µTUD, so that the open-circuit voltage almost applies on the load resistor. The inherent impedance of the µTUD and the load resistance is comparable in region II and the output voltage is sensitive to the load resistance. With the load resistance of 360 Ω, the µTUD reaches the peak output power of 54.4 nW.

To investigate the attenuation effect of tissues, a piece of porcine tissue (~30 mm thick) was inserted between the US transducer and the µTUD (Fig. 5b). In the same conditions, the dependence of the output voltage and output power on the load resistance is illustrated in Fig. 5d. It is observed that the maximal output voltage drops from 16.8 mV to 12.7 mV, while the peak output power drops from 54.4 nW to 21.2 nW with an addition of sound-attenuation medium. Therefore, the attenuation of bio-tissues is an indispensable factor in AET.

According to the limit set by FDA, the spatial-peak temporal-average intensity ($I_{SPTA}$) of ultrasound for continuous-wave application should be less than $720\,mW\,cm^{-2}$ for biological applications.

In the medium, the ultrasound intensity ($I$) of continuous wave is defined as

$$I = \frac{P^2}{2\rho c},\tag{7}$$

where $P$ is the amplitude of the ultrasound pressure, $\rho$ is the density of the medium, and $c$ is the sound speed in the medium. With the same experimental setup and settings, the µTUD was replaced by a hydrophone to measure the acoustic pressure at the location of the µTUD. The measured acoustic pressure amplitude ($P$) is 63 kPa and the calculated ultrasound intensity is $132\,mW\,cm^{-2}$. Because the output open-circuit voltage of the µTUD shows an approximately linear relationship with the incident acoustic pressure, the relationship between the output power on the load resistor and the incident acoustic pressure should also be linear. The peak output power on the load resistor can reach up to 297 nW if we give the incident acoustic pressure to $720\,mW\,cm^{-2}$.

So far, very few MEMS acoustic energy harvesters are reported for implanted devices. Supplementary Table 4 lists the published data of MEMS acoustic energy harvesters for implanted devices. It is noticed that the overall efficiencies are still low (generally lower than 1%). To power electronic devices, a high-frequency rectifier is also required. Although studies have reported the rectification circuit design and the efficiency can approach up to 89%[3], the threshold of a rectifier is usually hundreds of millivolts. Hence, the µTUD may not practically harvest energy for implanted device at current stage. However, an estimation of theoretically maximal efficiency of the µTUD is calculated in Supplementary Note 3, which can be up to 33%. It is reasonable to improve the µTUD's output voltage to 1–2 V, which is practically useful, by proper optimization in the future work.

Pulse experiments is also demonstrated to show the µTUD's potential of signal communication. With the similar experimental setup, burst-mode sinusoidal signals (10 $V_{p-p}$, 1 MHz) were input into the US transducer instead of continuous waves. A tee adapter was used to split the signal from the signal generator (Fig. 4a), so that both signals from the signal generator and the recovered signal from the µTUD can be monitored at the same time. In the oscilloscope, CH1 and CH2 give the signals from the signal generator and the recovered signal from the µTUD, respectively. Both signals were averaged for 64 times to stabilize the reading. Supplementary Figure 4 shows the relationship between the recovered signals and the cyclic number of the pulses. It is observed that the amplitude of the recovered signals keeps increasing until ~15 cycles. Then, the amplitude of the recovered signal almost does not change when further increasing cyclic number. Fig. 5e, f show the CH1 and CH2 signals, respectively,

when 20-cyc and 40-cyc sinusoidal pulse signals input. Within same time axis, ~56 µs traveling time of the ultrasound signals is obtained, indicating an ~85 mm separation. Based on the recovered signals, SNR can be calculated to be 20.54 dB. If the transmitted control signal is modulated by digits "0" and "1", so-called amplitude shift keying (ASK) modulation, digital information can be transmitted via the ultrasound link. In Fig. 5f, echo signals can also be seen for the US transducer. However, the echo signals dissipate significantly compared to the primary signals. Based on these experimental results, amplitude shift keying (ASK) modulation can be potentially adopted to carry information in the ultrasound signals. According to this demonstration, the modulated signals can be received by the µTUD via the ultrasound link and the signal communication can be realized.

**Further optimizations**. The power consumptions of most implanted devices are typically in the range of µW. Although the reported µTUD has demonstrated the potential of powering implanted devices, further optimizations should be performed to make it more practical. Integrated with MEMS technology, the optimization methods are proposed:

Firstly, the output power of TENG is highly dependent on the surface charge density. In the current design of the µTUD, the selection of silicon and silicon oxide is mainly because of the fabrication concerns. To meet the requirements of the fusion bonding, the output performance of the µTUD is partly sacrificed. Actually, most commonly used triboelectric materials are polymers, such as PTFE, polydimethylsiloxane (PDMS) and nylon. However, they are not compatible with the fusion bonding technique. An alternative is adhesive wafer bonding technique which can make a thin layer of polymer function as triboelectric layer. It is believed that this method is the most effective way to improve the output power of the µTUD. An estimation of theoretically maximal efficiency of the µTUD is provided in Supplementary Note 3, where the output power can be theoretically increased by 10,000 times.

Secondly, the effects of the µTUD's geometry on the output performance have not been comprehensively investigated yet. A brief discussion on µTUD's thickness and elastic modulus can be found in Supplementary Note 4. According to the discussion, a thinner and softer membrane will contribute to the higher energy conversion efficiency.

Thirdly, this work only focused on the design and fabrication of the µTUD and did not consider too much the electric circuit. In reality, energy loss in circuit is also an important concern, especially for the high-frequency circuit. A proper management circuit could also be designed to minimize this loss.

Last but not least, although MEMS technology can dramatically miniaturize the device size, biocompatibility should be carefully considered for implanted devices as well. Silicon has the drawback of instability in long-term in vivo application. One feasible solution to this issue is to add biocompatible coatings on the device. Previous research has already proven parylene-c and PDMS are biocompatible materials and compatible with MEMS fabrication[38]. By proper coating techniques, such as vapor deposition polymerization (VDP) and spin coating, a biocompatible thin layer can be formed atop the µTUD's surface. Since the thickness of the coating layer is controllable, we can still precisely design the desired device based on the modeling. Meanwhile, implanted devices are often complex systems, consisting of power units, control units, sensing units, etc. Hence, appropriate packaging strategies, including material and geometry, should be carefully designed for particular devices. A well-designed package can not only realize the biocompatibility, but also extremely decrease the acoustic attenuation.

## Discussion

For the very first time, a µTUD was developed based on the integration of TENG and MEMS process that significantly promotes the miniaturization and the integration level. Taking advantage of the wafer bonding process, the vacuum cavity was built to eliminate the effect of the ambient environment. AET through both oil and sound-attenuation medium were demonstrated. With the incident acoustic wave (63 kPa@1 MHz), the µTUD could achieve the open-circuit voltage of 16.8 mV$_{P-P}$ in oil. A linear dependence between the peak open-circuit voltage and the acoustic pressure was also observed. Under the FDA regulation, the reported µTUD can generate up to 297 nW output power. Although the current output power of the µTUD is low, optimization methods were proposed to improve the output and the theoretical analysis proves that the output power can be increased by 10,000 times. Among these methods, adhesive wafer bonding is expected as a promising way to fabricate the µTUD in the future. In addition, the potential of signal communication via ultrasound link was also demonstrated, which opens a new self-powered approach to communicate and control the wireless device.

## Methods

**Fabrication of the µTUD**. (i-ii) 300-nm-thick wet thermal silicon oxide was grown on a 100-mm P-doped 0.01 Ω cm <100> silicon wafer (500 µm thick). (iii) Next, the photolithography and reactive ion etching (RIE) process were performed to pattern the cavities. The depth of the cavity was set to 90 ± 10 nm. A standard clean process was conducted for the patterned wafer and an unpatterned SOI wafer (2 µm of P-doped 0.001 Ω cm device layer, 1 µm of boxing layer, 350 µm of handling layer) before fusion bonding. (iv) Then, fusion bonding process (Karl Suss, SB6e) was performed under the pressure of 3 kg cm$^{-2}$ and temperature of 480 °C in vacuum, followed by 1100 °C annealing. After this step, the cavities with vacuum of 10$^{-3}$ Pa were formed. (v) A combination of RIE and tetramethylammonium hydroxide (TMAH) etching was used to release the handling layer of the SOI wafer. The boxing layer was then removed by 5:1 buffered oxide etching (BOE) solution. (vi) Another photolithography and RIE process were conducted to expose the bottom silicon (as bottom electrode). (vii) Lastly, an e-beam deposition, followed by lift-off process, was done to create Ni (5 nm)/Au (150 nm) contact pads for wire-bonding.

**Characterizations of the µTUD**. A SKPM (Cypher S, Oxford Instruments) was performed to measure the surface potential distribution of the silicon oxide. A scanning electron microscope (SEM, Carl Zeiss Supra 55) was used to investigate the cross-sectional structure of the vacuum cavity. An optical microscope (Leica, DM4000M) was used to show the micro-structure of the µTUD. To measure the electrical output of the µTUD, an arbitrary function generator (RIGOL, DG4102) was used to drive the commercial US transducer (ABLL, China) while a digital storage oscilloscope (Agilent Technologies, DSO7104B) was adopted to measure the output voltage of the µTUD. An impedance analyzer (WK, 6500B) was utilized to characterize the commercial piezo transducer. A hydrophone system (Precision Acoustics, 46 mV MPa$^{-1}$ sensitivity @1 MHz) was used to calibrate the acoustic pressure in the vegetable oil.

## Data availability

The data that support the plots within this paper and other findings of this study are available from the corresponding author on reasonable request.

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

## Acknowledgements

The work is supported by National Natural Science Foundation of China (NSFC) (No. 61804103, U1932124), the National Science and Technology Major Project from Minister of Science and Technology of China (Grant No. 2018AAA0103104), National Key R&D Program of China (No. 2017YFA0205002), Natural Science Foundation of the Jiangsu Higher Education Institutions of China (No. 18KJA535001), Natural Science Foundation of Jiangsu Province of China (No. BK20170343), China Postdoctoral Science Foundation (No. 2017M610346), State Key Laboratory of Silicon Materials, Zhejiang University (No. SKL2018-03), Collaborative Innovation Center of Suzhou Nano Science & Technology, the Priority Academic Program Development of Jiangsu Higher Education Institutions (PAPD), the 111 Project and Joint International Research Laboratory of Carbon-Based Functional Materials and Devices. This work was also supported by the National Science and Engineering Research Council (NSERC) and Canada Research Chairs program (CRC).

## Author contributions

C.C., Z.W., X.H.S., and J.T.W.Y. conceived the idea, discussed the data and prepared the manuscript. C.C., X.J., and P.L. performed micro-fabrication of all structures and devices. C.C. X.H.S., and J.T.W.Y. led the mechanical modeling and theoretical studies. C.C. and Z.W. performed electrical measurements and analyzed the data. J.S. performed the SKPM measurements.

## Competing interests

The authors declare no competing interests.

## Additional information

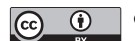

