## [Peer Review File · Nature Communications]

Reviewers' comments:

Reviewer #1 (Remarks to the Author):

This report on a micro triboelectric ultrasonic device is clearly an interesting and thorough examination of the topic. As far as I know, it is the first time a triboelectric device is completely fabricated based on MEMS technology, much improving its miniaturization and integration. And optimization methods were also proposed to improve the output. The fact that the device can demonstrate signal communication via ultrasound is certainly noteworthy. Overall, this manuscript should be taken into consideration of the publication in Nature Communications after clarifying the following comments.

1. The structure of the μ TEH is quite similar to that of capacitive micromachined ultrasonic transducer (CMUT). Please give some details about the difference between them regarding structure, working mechanism and fabrication process.
2. The operating mode the author is using for the μ TEHs is highly asymmetric since on the negative phase of the cycle there is contact between the SiO₂ and the Si and in the positive phase there is no contact. However, none of the measurements show any anharmonicity and appear quite sinusoidal (e.g. Figure 4c, 5e, 5f, S2). Please give some explanation.
3. The plot of Figure 4d seems fairly noisy. Please explain where the noise comes from.
4. The traveling time in Figure 5e/f and Figure S2 seems different. Please give some explanation.
5. Based on the acoustic experimental setup, the standing wave effect may also significantly affect the output of the device. I would like the author to give some explanation.

Reviewer #2 (Remarks to the Author):

The manuscript by Chen Chen et al. reports on a microtriboelectric energy harvester (μ TEH) developed to be applied for acoustic energy transfer and signal communication via an ultrasound link. The novelty of the paper mainly relies on the fabrication method which is based on silicon-based MEMS technologies. This approach extends to silicon MEMS the possibility to use the triboelectric effect, making it possible a better integration and miniaturization of implanted device. On the other end, however, it is not exploiting the main advantage of triboelectric μ TEH device which comes from the application of cheap and flexible polymeric materials and technologies.

It is a comprehensive study which is more interesting for the approach than for the results and their applicability. It could be of interest for the MEMS community but it is far from any real clinical application. I am not totally convinced that it is suited for Nature communication.

A few comments and questions:

- The cavity is under vacuum. This prevents some sort of squeeze dumping. It would be interesting to know the advantage of this approach over having atmospheric pressure on both sides of the membrane (with holes on it), or vacuum on both sides.
- There is a discrepancy between analytical calculation and COMSOL simulation which is attributed to the presence of the gold electrode in the comsol simulation only. The authors should state if comsol simulation without gold has been also performed and what is the result.
- There should be a trade-off between contact area, sticking force and membrane elastic modulus. Can the authors comment on how this affects EH efficiency?
- Figure 5b is totally unclear.
- The authors states that a rectifying efficiency of 89% is possible, citing a paper by Ozeri et al. However, based on the harvested power and voltage levels of the authors' technology (a few mVolts), the harvested energy can be practically zero. Indeed, voltage threshold of AC-DC rectification is typically around hundreds of mV. This must be better specified.
- What is the experimental efficiency achieved? Can you estimate the theoretical maximum transfer efficiency?

English syntax and grammar must be carefully checked throughout the manuscript (e.g. in line 146 "which 3D structure" must be "whose 3D structure"). In particular abstract and intro must be rewritten to better contextualize and describe the work.

On line 360 the degrees symbols are not well shown.

Reviewer #3 (Remarks to the Author):

1. This work is quite similar to the paper on Science by Prof Sang-Woo Kim (Science, 365. 6452, 491-494.), Comparing the output of the two articles, the device of Kim's team reached 6.74mW, which is 4 orders higher than this work (243 nW). With such an inferior performance, what is the advantage of the μ TEH on energy harvesting?
2. The μ TEH is designed for the implanted device, but there is no experiment proof to demonstrate the such application or its biocompatibility. Please consider such demonstrations to improve the manuscript.
3. The energy harvesting experiment was performed in oil of uniform density, but for an implanted device, the skin interface and muscle tissue can cause significant attenuation of ultrasound, more relevant experiments should be performed to verify the μ TEH's performance in sound-attenuating medium.
4. In Figure 4, the piezo transducer was actuated by a continuous wave with the amplitude of 10 V, why did the actual driving signal decrease to ~ 5 V? In addition, the manuscript does not explain how Figure 4b was tested, and the angle in Figure 4b is not explained.
5. The top electrode with charge should be given in the schematic of Figure 1a to better explain the working principle. In addition, what is the bottom electrode of Figure 3C? Is that silicon? How can it work as an electrode? Please provide a complete device schematic, including all materials (friction layer, electrode, etc.), and draw a charge transfer diagram of the TENG workflow.
- 6, In fig.1b-c, the KPFM data of the surface potential is poor and lacking proper level labeling. In addition, such a surface potential comparison cannot help support the conclusion without a reference like aluminum or PTFE.
- 7, It is interesting to find that the input impedance of such a TENG is below 1k ohm, which is different from the usual range of previously reported TENGs. Why is the input impedance so low? It is suggested the authors should also provide the corresponding output current data of their device.
- 8, It is interesting to see that the output power of this TENG device is still limited to nW level while the output voltage is up to several volts with low impedance. Why is the power output so low?

Response Letter

Reviewer #1:

This report on a micro triboelectric ultrasonic device is clearly an interesting and thorough examination of the topic. As far as I know, it is the first time a triboelectric device is completely fabricated based on MEMS technology, much improving its miniaturization and integration. And optimization methods were also proposed to improve the output. The fact that the device can demonstrate signal communication via ultrasound is certainly noteworthy. Overall, this manuscript should be taken into consideration of the publication in Nature Communications after clarifying the following comments.

1. The structure of the μ TEH is quite similar to that of capacitive micromachined ultrasonic transducer (CMUT). Please give some details about the difference between them regarding structure, working mechanism and fabrication process.

Response:

Thank the reviewer for this professional comment. We have drawn two different structural diagrams of the micro triboelectric ultrasonic device (μ TUD) and a capacitive micromachined ultrasonic transducer (CMUT), as illustrated in Figure R1. From the viewpoint of structure, the basic structure of the μ TUD and CMUT is quite similar. The main difference is the depth of the vacuum cavity. Usually, the depth of CMUT's cavity should be around 200 nm, while the μ TUD should limit the cavity depth less than 100 nm. The cavity depth is much more critical to the μ TUD. A typical cavity depth for CMUT is 200~300 nm, which is easy to be achieved by reactive ion etching (RIE). However, the cavity depth for the proposed μ TUD is less than 100

nm and requires much higher accuracy. Thus, the fabrication of the μ TUD is highly dependent on the precision of the RIE process.

For the working mechanism, conventional CMUT works under a DC bias (~80% of the collapse voltage) and the membrane should never touch the insulating layer (Figure R1a). With the incident ultrasonic wave, the vibrating membrane causes the change of CMUT's capacitance. Because of the constant DC bias, the capacitance change in the circuit will be reflected as current. Thus, CMUT needs continuous power supply to work properly. Regarding the μ TUD, the membrane and insulating layer should touch each other to make the triboelectrification occur (Figure R1b). The working mechanism of the μ TUD is the conjugation of triboelectrification and electrostatics. After the contact, the membrane and insulating layer will be oppositely charged. The vibrating membrane will cause a peak current in the external circuit. Therefore, μ TUD is a new kind of self-powered device and works without external electric power supply. These two kinds of devices work in very different ways. In practice, the DC bias for CMUT is often more than 100 V, which is a huge power consumption and causes safety concern.

Figure R1. Schematic illustration of basic structure design and working mechanism of (a) a capacitive micromachined ultrasonic transducer (CMUT) and (b) a micro triboelectric ultrasonic device (μ TUD).

2. The operating mode the author is using for the μ TEHs is highly asymmetric since on the negative phase of the cycle there is contact between the SiO_2 and the Si and in the positive phase there is no contact. However, none of the measurements show any anharmonicity and appear quite sinusoidal (e.g. Figure 4c, 5e, 5f, S2). Please give some explanation.

Response:

We highly appreciate the reviewer's professional comments. The measured output voltage is actually asymmetric in positive and negative phase. In fact, the operating mode of the μ TUD is indeed asymmetric. However, the displacement of the membrane is quite small and is comparable to the cavity depth. Thus, the response of the two half cycles only have tiny difference. It is not obvious in Figure 4b and S4 because the difference is really small compared to the peak value. Figure R2 is the output voltage curve of the μ TUD when $20\ \Omega$ load resistor is connected. We can clearly see that the output is asymmetric.

Figure R2. Output voltage curve of the μ TUD when connected with a $20\ \Omega$ load resistor.

3. The plot of Figure 4d seems fairly noisy. Please explain where the noise comes from.

Response:

Thanks for the reviewer's professional comment. It is because the separation between the ultrasound (US) transducer and the μ TUD is ~ 30 mm, which is within the near field of the piezo transducer. Inevitably, the generated acoustic pressure within the near field is relatively perturbative.

4. The traveling time in Figure 5e/f and Figure S2 seems different. Please give some explanation.

Response:

Thank the reviewer for your professional question. As we can see from two figures, echoes are also measured. If the separation between the US transducer and the μ TUD is too small, the main signals will be overlapped with echo signals. Therefore, we set the separation from 30 mm to 85 mm as shown in Figure 5 e/f, to clearly distinguish the recovered signals and the echo signals.

5. Based on the acoustic experimental setup, the standing wave effect may also significantly affect the output of the device. I would like the author to give some explanation.

Response:

We highly appreciate the reviewer's professional comment. The standing wave effect does affect the output of the device. In the experiments, the variation of the output power along the axial direction can be observed. According to this comment, we have added some description in the manuscript (page 12), as following:

“In addition, the standing wave effect also affect the output of the device. The variation of the output voltage along the axial direction was observed (Figure 4d). This phenomenon is due to the standing wave effect (Supplementary Note S2). Since US transducer and the μ TUD are well

aligned in the acoustic energy transfer, the standing wave may be generated between the transmitter and receiver. Considering the acoustic impedance mismatch between transducers and medium, acoustic wave reflects every time it hits the transducer. These reflections make the acoustic energy persistent in the form of the standing wave. Therefore, the efficiency of the ultrasonic link will be sensitive to the separation between transmitter and receiver. Only when the separation is integer multiples of one-half wavelength, the efficiency of the ultrasound link will be maximized.”

Figure 4. (d) Standing wave effect in the experiments.

Reviewer #2:

The manuscript by Chen Chen *et al.* reports on a microtriboelectric energy harvester (μ TEH) developed to be applied for acoustic energy transfer and signal communication *via* an ultrasound link. The novelty of the paper mainly relies on the fabrication method which is based on silicon-based MEMS technologies. This approach extends to silicon MEMS the possibility to use the triboelectric effect, making it possible a better integration and miniaturization of implanted device. On the other end, however, it is not exploiting the main advantage of triboelectric μ TEH device which comes from the application of cheap and flexible polymeric materials and technologies. It is a comprehensive study which is more interesting for the approach than for the results and their applicability. It could be of interest for the MEMS community but it is far from any real clinical application. I am not totally convinced that it is suited for Nature communication. A few comments and questions:

1. The cavity is under vacuum. This prevents some sort of squeeze damping. It would be interesting to know the advantage of this approach over having atmospheric pressure on both sides of the membrane (with holes on it), or vacuum on both sides.

Response:

We highly appreciate the reviewer's professional comment. Other than the squeeze damping, the application is the most important reason to choose this sealed vacuum structure. In this work, the μ TUD is designed for in-immersion application, fluid (such as bodyfluid) would get into the cavity if it is not fully sealed.

As far as we know, holes on the membrane cannot reduce the damping. The fluid (even the air) passing through the holes will cause viscous and thermal loss. This energy loss is much larger than other types of energy loss, resulting in a significant damping.

Actually, at the early stage of design, vacuum on both sides was an option for us. However, we failed to run it because ultrasound is a mechanical wave and the propagation of ultrasound needs medium. The ultrasonic energy would not be transmitted to the membrane if it is vacuum in front of the membrane. In our future work, we will think about whether there is a solution to this problem.

2. There is a discrepancy between analytical calculation and COMSOL simulation which is attributed to the presence of the gold electrode in the COMSOL simulation only. The authors should state if COMSOL simulation without gold has been also performed and what is the result.

Response:

According to the reviewer’s suggestion, we have added the COMSOL simulation results where gold electrode was ignored in the revised manuscript as Table S3. The simulated results perfectly match the results of analytical model. Detailed discussion can be found in the revised manuscript (page 9), as following:

Table S3. Results of the theoretical analysis

		Static Displacement	Resonant Frequency
(Analytical)		79.4 nm	1.18 MHz
(Numerical)	With top gold layer	74.5 nm	1.17 MHz
	Without top gold layer	79.1 nm	1.19 MHz

“The results of the theoretical analysis are presented in Table S3. The resonant frequencies of the membrane are 1.18 MHz and 1.19 MHz from analytical modeling and numerical analysis, respectively. It should be noted that the resonant frequency will slightly increase if we eliminate

the top gold layer. This discrepancy is because the addition of the top gold layer decreases the effective Young's modulus of the membrane.”

“Based on the results of the analytical modeling, 3D mapping of the static displacement surface was plotted for the membrane as shown in Figure 2b. Meanwhile, the static displacement surface of the membrane under the atmospheric pressure was simulated by COMSOL and the result is illustrated in Figure 2d. Due to the computational capacity, only one quarter geometry was built and symmetric conditions were applied. Ignoring the top gold layer, the maximal displacements of the membrane under the atmospheric pressure are respectively 79.4 nm and 79.1 nm *via* two methods.”

3. There should be a trade-off between contact area, sticking force and membrane elastic modulus. Can the authors comment on how this affects EH efficiency?

Response:

Thank the reviewer for this question. Increasing contact area can definitely improve the EH efficiency because more triboelectric charges are transferred between two materials. We investigated how the membrane's thickness and Young's modulus affect the contact area, and found that a thinner and softer membrane will give larger contact area. Furthermore, potential sticking problem was also thoroughly discussed, especially electrostatic force. An estimation of theoretically maximum EH efficiency was also obtained based on this discussion. Detailed discussion has been listed in the Supplementary Note S4, as following:

“Note S4. Mechanical parameters of the μ TUD

Both material and structure factors may affect the performance of the μ TUD. A brief discussion will be given here to guide the design of the μ TUD in the future.

Mechanically, the μ TUD may fail to work due to the collapse of membrane (stick to the substrate). Because of the wafer bonding technique, there is no wet etching step for cavity formation so that the capillary adhesion is avoided. Moreover, the silicon membrane is the device layer (single crystalline silicon) of SOI wafer and has a very low intrinsic stress. The stress factor can also be ignored in this discussion. Therefore, the main factor contributing to the failure of the μ TUD is electrostatic force, induced by the triboelectric charges. A more detailed discussion about the electrostatic force has been provided in Note S3.

Refer to the definition of performance figure-of-merit, the material's triboelectric properties play the most important role in determining TENG's output. The improvement of surface charge density will dramatically increase TENG's output power. With same triboelectric materials, increasing the contact area can effectively improve the device's efficiency because more triboelectric charges are transferred between two materials. Numerical analysis (COMSOL) was carried out to investigate the effect of membrane's thickness and Young's modulus, respectively.

To investigate effect of the membrane's thickness, the Young's modulus was set to 170 GPa. Several combinations of membrane parameters were selected to keep the consistent resonant frequency (as shown in Table S5). With same incident acoustic pressure, the displacement responses of the membranes were numerically simulated and the results are shown in Figure S6a. It is seen that a thinner membrane gives a larger dynamic displacement in the same conditions. For a given separation between the membrane and the substrate, the larger dynamic displacement can result in a larger contact area.

To investigate the effect of membrane's Young's modulus, the thickness of the membrane was set to 2 μ m. Another group of parameter combination were selected, which is shown in Table S6. With same incident acoustic pressure, the displacement responses of the membranes

were numerically simulated and the results are shown in Figure S6b. It is observed that membrane with the lower Young's modulus has a larger dynamic displacement so that a larger contact area can be achieved.

In conclusion, a thinner and softer membrane can contribute to a larger contact area, resulting in a higher efficiency.”

Figure S6. Dynamic displacement of membranes. Dynamic displacement of membranes with (a) different thickness and (b) different Young's modulus.

4. Figure 5b is totally unclear.

Response:

Thank the reviewer for the kind reminder. The figure has been modified carefully and the revised experimental photographs have been demonstrated in Figure 5a and 5b.

Figure 5. Photograph of the ultrasonic link (a) without and (b) with sound-attenuation medium (porcine tissue).

5. The authors states that a rectifying efficiency of 89% is possible, citing a paper by Ozeri *et al.* However, based on the harvested power and voltage levels of the authors' technology (a few mVolts), the harvested energy can be practically zero. Indeed, voltage threshold of AC-DC rectification is typically around hundreds of mV. This must be better specified.

Response:

Thank the reviewer for the constructive comment. We revised the description and specified the threshold problem in the manuscript (page 15), as following:

“So far, very few MEMS acoustic energy harvesters are reported for implanted devices. Table S3 lists the published data of MEMS acoustic energy harvesters for implanted devices. It is noticed that the overall efficiencies are still low (generally lower than 1%). To power electronic devices, a high-frequency rectifier is also required. Although studies have reported the rectification circuit design and the efficiency can approach up to 89%, the threshold of a rectifier is usually hundreds of millivolts. Hence, the μTUD may cannot practically harvest energy for implanted device at current stage. However, an estimation of theoretically maximal efficiency of the μTUD is calculated in Note S3, which can be up to 33%. It is reasonable to improve the

μ TUD's output voltage to 1~2 V, which is practically useful, by proper optimization in the future.”

6. What is the experimental efficiency achieved? Can you estimate the theoretical maximum transfer efficiency?

Response:

Thank the reviewer for the professional questions. The experimental efficiency of the μ TUD is around 2×10^{-4} % through sound-attenuation medium, as presented in Table S3. This low efficiency is due to the low surface charge density. Currently, we used silicon and silicon oxide as triboelectric pair. Fabrication is a very important concern for this material selection, which sacrifices the energy conversion efficiency of the μ TUD. Secondly, the actual contact area is small compared to the membrane size. Thus, triboelectric charges are not sufficiently transferred between two materials.

An estimation has been carried out in Supplementary Note S3 to calculate the theoretically maximum transfer efficiency. We assumed that the whole membrane is fully contact with the substrate and calculated the theoretically maximum surface charge density. Based on the analysis, the efficiency can be up to ~33% and the output of the μ TUD can be potentially increased by 10,000 times. The detailed calculation is as follows:

“Note S3. An estimation of the theoretically maximum efficiency

The electrostatic force, induced by triboelectric charges, will be the main reason to make the membrane collapse. Let's assume an extreme case: After sufficient electrification, the accumulated charges establish an electric field which attracts the membrane to bend. There is a

point where electrostatic force is too strong to be balanced by the membrane's restoring force. After this point, the membrane will collapse and the device will fail to work.

A μ TUD cell (gold layer is ignored) is modelled as shown in the figure and mass-spring-damping model is used to analyze its behavior. Here, we assume the total pressure (induced by atmospheric pressure and electric field) on the membrane as P . Refer to Note S1, the membrane's deflection is defined as

$$u(r) = \frac{Pa^4}{64D} \left(1 - \frac{r^2}{a^2}\right)^2 \quad (11)$$

where a is the radius of the membrane and r is the distance from the center.

The maximal deflection of the membrane is at $r=0$:

$$u_{\max} = \frac{Pa^4}{64D} \quad (12)$$

The average deflection of the membrane is defined as

$$u_{\text{avg}} = \frac{\int_0^a 2\pi r u(r) dr}{\pi a^2} = \frac{Pa^4}{192D} = \frac{u_{\max}}{3} \quad (13)$$

In the case of small deflection (much smaller than the membrane's thickness), linear system is assumed and the restoring force exerted on the membrane can be written in terms of the average deflection and a linear equivalent spring constant (k):

$$F_m = k \cdot u_{\text{avg}} \quad (14)$$

Deriving from $P\pi a^2 = k \cdot u_{\text{avg}}$, the equivalent spring constant is obtained as

$$k = \frac{\pi a^2 P}{u_{\text{avg}}} = \frac{192\pi D}{a^2} \quad (15)$$

To simplify the calculation, we assume the μTUD as a parallel plate capacitor and two contact surfaces are uniformly charged. The critical voltage is defined as

$$V_{PI} = \sqrt{\frac{8}{27} \frac{k \left(g_0 - \frac{\pi a^2 p_0}{k} \right)^3}{\varepsilon_0 \pi a^2}} \quad (16)$$

where ε_0 is the vacuum permittivity, g_0 is the depth of cavity and p_0 is the atmospheric pressure.

Substituting the parameters in Table S2, we can obtain this critical voltage $V_{PI}=5.69$ V.

Hence, the maximal effective surface charge density (σ_{\max}) is

$$\sigma_{\max} = \frac{V_{PI} \cdot \frac{\varepsilon_0 \pi a^2}{3g_0}}{\pi a^2} = \frac{3\varepsilon_0 V_{PI}}{2g_0} = 840 \mu\text{C}/\text{m}^2 \quad (17)$$

Therefore, as long as the effective surface charge density is less than $840 \mu\text{C}/\text{m}^2$, the μTUD can work properly.

According to the reference, the open circuit of the μTUD can be defined as

$$V_{OC} = \frac{d\sigma \cdot 2g_0}{\varepsilon_0(d+2\varepsilon_r g_0)} \quad (18)$$

where d is the thickness of the silicon oxide layer and ε_r is the relative permittivity of the silicon oxide layer. To simplify the model, we assume the vibration of the membrane is symmetric, so that the maximal separation is $2g_0$. The equation can be rewritten as

$$\sigma = \frac{V_{OC}\varepsilon_0(d+2\varepsilon_r g_0)}{2g_0 d} \quad (19)$$

Based on our experimental results, the open-circuit voltage of the μTUD is 16.8 mV. The actual effective surface charge density is calculated as $2.71 \mu\text{C}/\text{m}^2$. There are two reasons for this low surface charge density currently. Firstly, the selection of silicon and silicon oxide sacrifices some performance of the μTUD . Secondly, the actual contact area is small compared to the membrane size. Thus, triboelectric charges are not sufficiently transferred between two materials. Based on Equation (18),

$$V_{oc} \propto \sigma$$

So that output power (P) has the relation:

$$P \propto \sigma^2$$

Hence, the energy harvesting efficiency of the μ TUD can be theoretically increased by more than 10000 times, reaching ~33%.

Although it is extremely difficult to practically achieve this high energy conversion efficiency, it is still reasonable to achieve a 1~2 V output voltage by proper optimization method, such as adding a charge trapping structure or additional triboelectric layer.”

7. English syntax and grammar must be carefully checked throughout the manuscript (*e.g.* in line 146 “which 3D structure” must be “whose 3D structure”). In particular abstract and intro must be rewritten to better contextualize and describe the work.

Response:

We have carefully checked the syntax and grammar and revised the manuscript accordingly in the revised version.

8. On line 360 the degrees symbols are not well shown.

Response:

Thank reviewer for the helpful comment and suggestion. Please check the revised manuscript (page 18), as following:

“Then, fusion bonding process (Karl Suss, SB6e) was performed under the pressure of 3 kg/cm² and temperature of 480°C in vacuum, followed by 1,100°C annealing. After this step, the cavities with vacuum of 10⁻³ Pa were formed.”

Reviewer #3:

1. This work is quite similar to the paper on Science by Prof. Sang-Woo Kim (Science, 365. 6452, 491-494.), comparing the output of the two articles, the device of Kim's team reached 6.74 mW, which is 4 orders higher than this work (243 nW). With such an inferior performance, what is the advantage of the μ TEH on energy harvesting?

Response:

Thank the reviewer for this professional comment. Sang-Woo Kim's group developed a capacitive triboelectric energy harvester to capture ultrasound energy and power implanted device. The device used commonly-used materials and achieved a high efficiency for transcutaneous ultrasound energy transfer in centimetre size. It is a pioneering and great work to push the triboelectric nanogenerator forward in vivo applications and to power medical implants.

The best novelty of this demonstrated work is the integration of the triboelectric device and MEMS technology. This is the very first MEMS triboelectric device in the world. Based on MEMS technology, we introduced a new method for design of microstructured triboelectric device working at a desired frequency. We also developed a fabrication process which is compatible with batch fabrication process. The concepts and the techniques proposed in the manuscript provided a new perspective for designing and fabricating triboelectric device in the future.

Although the demonstrated performance is currently inferior compared to the work of Prof. Sang-Woo Kim's group (Science, 365. 6452, 491-494), the output of the μ TUD can be potentially increased by 10,000 times according to the theoretical analysis (Supplementary Note S3).

To more clearly describe novelty of this work, we have carefully revised the abstract, introduction and conclusions in the revised manuscript.

2. The μ TEH is designed for the implanted device, but there is no experiment proof to demonstrate the such application or its biocompatibility. Please consider such demonstrations to improve the manuscript.

Response:

Thank reviewer for the helpful suggestion. An *in-vitro* experiment has been implemented to demonstrate the performance of the μ TUD in the sound-attenuating medium. Please see the revised manuscript for more details (page 14), as following:

“To investigate the attenuation effect of tissues, a piece of porcine tissue (~30 mm thick) was inserted between the US transducer and the μ TUD (as shown in Figure 5b). In the same conditions, the dependence of the output voltage and output power on the load resistance is illustrated in Figure 5d. It is observed that the maximal output voltage drops from 16.8 mV to 12.7 mV, while the peak output power drops from 54.4 nW to 21.2 nW with an addition of sound-attenuation medium. Therefore, the attenuation of bio-tissues is an indispensable factor in acoustic energy transfer.”

3. The energy harvesting experiment was performed in oil of uniform density, but for an implanted device, the skin interface and muscle tissue can cause significant attenuation of ultrasound, more relevant experiments should be performed to verify the μ TEH's performance in sound-attenuating medium.

Response:

Thank the reviewer for this professional comment. We supplemented an *in-vitro* experiment to demonstrate the application of powering implanted device. Porcine tissue was used as the sound-attenuation medium between the US transducer and the μ TUD. Related discussion has also been added in the revised manuscript (page 14).

4. In Figure 4, the piezo transducer was actuated by a continuous wave with the amplitude of 10 V, why did the actual driving signal decrease to ~ 5 V? In addition, the manuscript does not explain how Figure 4b was tested, and the angle in Figure 4b is not explained.

Response:

Thank the reviewer for the kind reminder. The impedance analysis in Figure S3 (revised version) determines the selection of the signal generator's impedance mode. It can be seen that the impedance amplitude of the piezo transducer is 57.4Ω at the frequency of 1 MHz. It is close to the output impedance of the signal generator ($\sim 50 \Omega$). The equivalent circuit is shown in Figure R3 when the piezo transducer is connected to the signal generator. The signal generator can be considered as a series of ideal voltage source and output impedance. The actual driving voltage on the piezo transducer (V_{load}) is $V_{load} = V_0 \cdot \frac{Z_{load}}{Z_{output} + Z_{load}}$, which is approximately half of the source voltage.

Figure R3. Equivalent circuit when the piezo transducer is connected to the signal generator.

This confusion of the driving signal is due to the unclear description in the manuscript. Actually, both the “10 V” and “5 V” are the measured values by oscilloscope (1 M Ω input impedance). If we set the impedance as “50 Ω ” mode in signal generator, the read on the signal generator is almost the actual driving signal on the US transducer. We also revised the description in the manuscript (page 12) to avoid the confusion, as following:

“During the experiments, the US transducer was actuated by a continuous wave with the amplitude of 5 V and the frequency of 1 MHz.”

We also removed the angle line in the impedance analysis figure because it is irrelevant to this experiment.

5. The top electrode with charge should be given in the schematic of Figure 1a to better explain the working principle. In addition, what is the bottom electrode of Figure 3C? Is that silicon? How can it work as an electrode? Please provide a complete device schematic, including all materials (friction layer, electrode, *etc.*), and draw a charge transfer diagram of the TENG workflow.

Response:

We highly appreciate the reviewer’s professional questions and suggestions. Actually, both the silicon wafer and the device layer of the SOI wafer are highly conductive silicon (less than 0.01 ohm-cm in resistivity). Thus, the bottom silicon layer does act as bottom electrode. The silicon oxide works as a pure friction layer. Regarding the silicon membrane and the top gold layer, on the one hand, the silicon membrane functions as another friction layer. On the other hand, the combination of silicon membrane and gold layer functions as top electrode. The charges will flow through silicon membrane, top gold layer to external circuit when the

membrane vibrates. Therefore, the μ TUD is working under a conductor-dielectric contact-separation mode. In addition, the gold layers on both the bottom silicon layer and the membrane are also necessary for wire bonding. To state this issue clearly, a better schematic of structure design and working mechanism is drawn in Figure 1 in revised version.

Figure 1 | Structure design and working mechanism of the μ TUD. (a) Representative diagram of the μ TUD. (b) Exploded view structure of the μ TUD. (c) Schematic of the working mechanism of the μ TUD during vibration.

We also revised the manuscript to explain the working mechanism more clearly (page 5-6), as following:

“Essentially, the basic structure of the μ TUD is a micro capacitor, where a fully clamped silicon membrane is suspended atop a vacuum cavity (Figure 1a). Highly-doped silicon layer underneath the silicon oxide (triboelectric & dielectric layer) acts as bottom electrode. The silicone membrane is also highly conductive so that the combination of silicon membrane and top gold layer functions as top electrode. Figure 1b and 1c illustrate the basic structure and the working mode (conductor-dielectric contact-separation mode) of the μ TUD. The suspended silicon membrane vibrates and contacts the silicon oxide layer under the excitation of the incident ultrasound. Then, the contact between silicon and silicon oxide causes the charge redistribution by triboelectrification.”

6. In fig.1b-c, the KPFM data of the surface potential is poor and lacking proper level labeling. In addition, such a surface potential comparison cannot help support the conclusion without a reference like aluminum or PTFE.

Response:

Thank the reviewer for this professional comment. We have conducted the KPFM experiments again and compared the results of SiO₂ and PTFE in Supplementary Information as **Figure S1**.

Figure S1. SKPM measurements. Surface potential distribution of SiO₂ (a) before contact electrification, (b) after contact with silicon, and (c) after contact with PTFE.

In addition, some discussion has also been revised in the manuscript, as following:

“Scanning Kelvin probe microscopy (SKPM) was performed on the surface of silicon oxide to obtain the surface potentials. The surface potentials before and after contacting are shown in Figure S1a and c. The surface potential after contact with Polytetrafluoroethylene (PTFE) film was also measured as a reference (Figure S1b). From the results, the surface potential of silicon oxide presents obvious increase after contact, indicating an accumulation of positive charges. When this contact-separation process repeats, AC power is generated in the external circuit. Although more significant electrification between PTFE and silicon oxide is noticed, trade-off should be made between the output performance and fabrication limit.”

7. It is interesting to find that the input impedance of such a TENG is below 1k ohm, which is different from the usual range of previously reported TENGs. Why is the input impedance so low? It is suggested the authors should also provide the corresponding output current data of their device.

Response:

Thank the reviewer for this comment. It should be noted that both device itself and working frequency can affect the input impedance.

Figure R4. Equivalent circuit of a typical TENG with load.

Figure R4 shows the equivalent circuit of a typical TENG with load. In the circuit, the TENG can be considered as a series of an ideal voltage source and a time-varying capacitor. Hence, the inherent impedance of the TENG is mostly contributed by the capacitor. The capacitive impedance of a capacitor is defined as:

$$X_c = \frac{1}{\omega C} = \frac{1}{2\pi f C}$$

where f is the frequency in Hertz and C is the capacitance in Farads.

We can find that the working frequency (f) plays a very important role in determining the capacitive impedance. Currently, the most majority of reported TENGs work under a very low frequency, typically under 100 Hz. The corresponding inherent impedance of these TENGs are usually in the range of mega Ohms. However, our μ TUD is working under a frequency of 1 MHz which is much higher than the conventional TENGs'. Refer to equation above, the inherent impedance of the μ TUD should be much lower.

We tried to measure the working current of the μ TUD. However, we cannot get precise results due to the limit of the equipment's resolution. Indirect evidence can support our experimental results in some extent. The structure of the μ TUD is inspired by capacitive micromachined ultrasonic transducer (CMUT). Regarding structure, the only difference between

the μ TUD and CMUT is the cavity depth. Typically, the working frequency of CMUT is several MHz. Many references (Ergun *et al*, 2003, 16, 76-84; Huang *et al*, 2003, 12, 128-137; Midtbo *et al*, 2006, 938-941; Zheng *et al*, 2019, 286, 39-45) have demonstrated that the input impedance of CMUT, at working frequency, ranges from tens of Ohms to hundreds of Ohms.

8. It is interesting to see that the output power of this TENG device is still limited to nW level while the output voltage is up to several volts with low impedance. Why is the power output so low?

Response:

We highly appreciate the reviewer’s professional question. We think the reviewer may mistake the unit of the output voltage in Figure 5. In the manuscript, the output voltage of the μ TUD is several millivolts, not volts. In Figure 5e and f, CH1 is the driving signals.

Figure 5. Input signals to US transducer and the output signal from the μ TUD driven by (e) 20-cyc and (f) 40-cyc sine waves (1 MHz, ~85 mm separation in oil).

Reviewers' comments:

Reviewer #1 (Remarks to the Author):

This revised manuscript clearly addressed the current limits of TENG technology in area of micro-/nano-electronics: miniaturization and chip-sized integration. The presented work focused on the integration of TENG and MEMS technology and demonstrated a good attempt to build a triboelectric device completely by MEMS process. They provided a solution to the addressed issues by proposing a comprehensive structure design method and a feasible fabrication approach. With the fabricated micro-triboelectric device, they characterized its capability of capturing ultrasound energy through oil and sound-attenuation medium and data communication. Although the power output is relatively low at present stage, this work is still an important advance in both fields of TENG and MEMS. In particular, the integration with MEMS technology may help TENG find a mass production way with standard consistency to the market. Last but not least, the authors also gave a reasonable estimation of the theoretically maximum output voltage/power for the micro-triboelectric device in the revised version. The calculated value shows the micro-triboelectric device does have huge potential for harvesting ultrasound energy. Overall, I would strongly recommend this revised manuscript for publication in Nature Communications

Reviewer #2 (Remarks to the Author):

The authors have satisfactorily answered to all my questions and concerns.

Reviewer #3 (Remarks to the Author):

1. Although the authors claim that the best novelty of this work is the integration of the triboelectric device and MEMS technology, I still recommend that the author make more experimental attempts to increase the output power of the device. Because the output of the device in this work is too low to achieve the energy harvesting application.
2. I don't agree with the author's answer to question 4 I raised. According to the author's answer, the actual driving voltage on the piezo transducer (V_{load}) is approximately half of the source voltage, which means the resistance of the piezoelectric transducer is equal to the output resistance of the signal generator (i.e., 50 ohm). But actually, the resistance of the piezoelectric transducer is much higher than 50 ohms. This must be better specified.
3. For implantable devices, biocompatibility is very important. The authors should provide evidence for the biocompatibility of silicon-based devices and how related processing if needed, would affect the device performance.
4. Since the device in this work can't power electronic equipment, some sentences should be modified, such as in the abstract, "Accordingly, a prototyped acoustic energy transfer system was demonstrated for powering implanted devices." Powering implanted devices has not been demonstrated. The authors should review the manuscript, reorganize the abstract, introduction, and conclusion sections.

Response Letter

Reviewer #3:

1. Although the authors claim that the best novelty of this work is the integration of the triboelectric device and MEMS technology, I still recommend that the author make more experimental attempts to increase the output power of the device. Because the output of the device in this work is too low to achieve the energy harvesting application.

Response: Thank the reviewer for this suggestion. Since 2012, TENG has experienced a very rapid development period. Currently, the urgent requirements for TENG based self-powered sensors are the miniaturization and high-accuracy. This work demonstrated an acoustic energy transfer and signal communication device, rather than an energy harvester. By coupling the technologies of TENG and MEMS, we set a world record to fabricate the smallest triboelectric device. With 63 kPa@1 MHz ultrasound input, the fabricated μ TUD can generate the voltage signal of 16.8 mV and 12.7 mV through oil and sound-attenuation medium, respectively. Particularly, a relatively high signal-to-noise ratio (20.54 dB) can be achieved.

In order to make the focus of the manuscript clearer, we have modified the abstract, introduction, some discussion and conclusion in the revised manuscript, as follows:

Abstract: “Accordingly, a prototyped acoustic energy transfer system was demonstrated based on the reported μ TUD.” “With 63 kPa@1 MHz ultrasound input, the μ TUD can generate the voltage signal of 16.8 mV and 12.7 mV through oil and sound-attenuation medium, respectively.”

Introduction: “Acoustic energy transfer (AET) has becoming an attracting topic in low-power energy transfer.” “Several efforts have been devoted to developing self-powered triboelectric acoustic sensor (TAS)” “In this work, a microstructured triboelectric ultrasonic device (μ TUD)

was developed based on the coupling technologies of TENG and MEMS.” “Moreover, AET through oil and sound-attenuation medium were demonstrated and 16.8 mV and 12.7 mV output voltage were achieved respectively.”

Conclusion: “For the very first time, a μ TUD was developed based on the integration of TENG and MEMS process that significantly promotes the miniaturization and the integration level.”

For all that, we still consider the suggestion from the reviewer is very valuable and desirable. Thus, in future work, we will continue to improve the output performance of μ TUD. The detailed optimization approaches have also been proposed in the “Further optimizations” part.

2. I don't agree with the author's answer to question 4 I raised. According to the author's answer, the actual driving voltage on the piezo transducer (V_{load}) is approximately half of the source voltage, which means the resistance of the piezoelectric transducer is equal to the output resistance of the signal generator (*i.e.*, 50 ohm). But actually, the resistance of the piezoelectric transducer is much higher than 50 ohms. This must be better specified.

Response: Thank the reviewer for this professorial comment. It is true that the resistance of the piezoelectric transducer is much higher than 50 ohms. However, our system is an AC circuit, which means that not only resistance should be considered, but also other effects, such as reactance. Reactance is related to the frequency of the AC current. Therefore, we should use *impedance* to do the matching in AC circuit, instead of *resistance*. It should be noted that the frequency of the AC signal is 1 MHz. Although the resistance of the piezoelectric transducer is high, the amplitude of the impedance is low at this frequency. Figure S3 shows the impedance analysis of the piezoelectric transducer, which can prove our statement.

Figure S3. Impedance analysis of the commercial ultrasound transducer.

3. For implantable devices, biocompatibility is very important. The authors should provide evidence for the biocompatibility of silicon- based devices and how related processing if needed, would affect the device performance.

Response: Thank the reviewer for this professorial comment. According to this comment, we have added some description in the manuscript (page 17), as following: “Last but not least, although MEMS technology can dramatically miniaturize the device size, biocompatibility should be carefully considered for implanted devices. Silicon has the drawback of instability in long-term in vivo application. One feasible solution to this issue is to add biocompatible coatings on the device. Previous research has already proven parylene-c and PDMS are biocompatible materials and compatible with MEMS fabrication. By proper coating techniques, such as vapor deposition polymerization (VDP) and spin coating, a biocompatible thin layer can be formed atop the μ TUD’s surface. Since the thickness of the coating layer is controllable, we can still precisely design the desired device based on the modeling. Meanwhile, implanted devices are often complex systems, consisting of power units, control units, sensing units, etc. Hence, appropriate packaging strategies, including material and geometry, should be carefully designed for particular devices. A well-

designed package can not only realize the biocompatibility, but also extremely decrease the acoustic attenuation.”

4. Since the device in this work can't power electronic equipment, some sentences should be modified, such as in the abstract, “Accordingly, a prototyped acoustic energy transfer system was demonstrated for powering implanted devices.” Powering implanted devices has not been demonstrated. The authors should review the manuscript, reorganize the abstract, introduction, and conclusion sections.

Response: We really appreciate the reviewer for these kind suggestions. For the abstract, we have revised it carefully, as follows: “Accordingly, a prototyped acoustic energy transfer system was demonstrated based on the reported μ TUD. With 63 kPa@1 MHz ultrasound input, the μ TUD can generate the voltage signal of 16.8 mV and 12.7 mV through oil and sound-attenuation medium, respectively.” We have deleted the diction about “powering implanted devices” in the entire manuscript.

REVIEWERS' COMMENTS:

Reviewer #3 (Remarks to the Author):

The authors have properly addressed my comments. I recommend acceptance of this paper for publication.

Response Letter

REVIEWERS' COMMENTS

Reviewer #3 (Remarks to the Author)

The authors have properly addressed my comments. I recommend acceptance of this paper for publication.

Response: We appreciate the reviewer' positive comments and recognition on our work.